# Improving Synchronous Dataflow Analysis Supported by Petri Net Mappings

**José-Inácio Rocha [1,2,]\*, Octávio Páscoa Dias [3] and Luís Gomes [2,4]**

[1] Escola Superior de Tecnologia de Setúbal, Instituto Politécnico de Setúbal, Campus do IPS, 2914-761 Estefanilha, Portugal

[2] Center of Technology and Systems, UNINOVA-CTS, FCT Campus, 2829-516 Caparica, Portugal; lugo@fct.unl.pt

[3] Lusíada University of Lisbon, Rua da Junqueira, 188-198, 1349-001 Lisboa, Portugal; octavio.pdias@gmail.com

[4] Faculty of Science and Technology (FCT NOVA), Campus Caparica, 2829-516 Caparica, Portugal

\* Correspondence: jose.rocha@estsetubal.ips.pt; Tel.: +351-265-790-000

**Abstract:** Whereas most of the work that analyses Synchronous Dataflow (SDF) stays in the dataflow framework, this work pushes its analysis into another framework level, thereby addressing issues that are not well addressed or are even unexplored in SDF. In this manner, the paper proposes a model-driven engineering (MDE) method, combining Synchronous Dataflow (SDF) and Petri nets, to highlight and reinforce their interoperability in digital signal processing applications, cyber-physical systems, or industrial applications. Improvements regarding the settlement and exploitation of the initial conditions associated with SDF are demonstrated; this issue is crucial for every cyber-physical system, since a system's initial conditions are crucial to ensuring the system's liveness. The improvements outlined in this work exploit an innovating mapping in the Place/Transition (P/T) Petri net domain that is intended to reduce and predict the total amount of initial data in SDF channels. The relevance of the firing semantics engaged with the equivalent Petri net model is discussed. This paper proposes a new approach to estimate whether an SDF has a static schedule by performing simulation and property verification of the equivalent-based P/T Petri net system achieved, framed by a Petri net invariant analysis and based on the stubborn set method of Petri nets. In this way, this new approach will allow mitigating the state explosion problem. Finally, a strategy is applied to two case studies to discover all the elementary circuits (static schedules) associated with the generated model's state-space.

**Keywords:** synchronous dataflow; Petri net; digital signal processing computation; place and transition invariant; multi-rate dataflow

## 1. Introduction

Signal processing-based applications typically demand initial conditions to start a model's execution, which can also be associated with the system's initialization and configuration. In this paper, this period is referred to as the start-up phase. In some cases, its evolution lasts for a significant amount of time. In this context (associated with regular execution), one must be aware that before the regular execution sequence, a start-up phase will always take place, and its length is not or cannot either be easily bounded or predictable. The major goal of the proposed work is to find initial markings such that system's model enjoys liveness and consistency properties, in addition to minimizing the runtime and reducing the buffer requirement in the system start-up phase. This strategy potentially will have a strong impact on reducing a system's energy consumption.

Currently, the electronic industry is pushed into embedded solutions with a multitude of powerful features integrating software and hardware processor engines. Moreover, the imposed constraints on timing and energy pushed the design problem into another stage: model-driven methods [1].

To describe the behavior of complex systems, the designers use Model-Driven Engineering (MDE) methods. The goals of this approach are to obtain the benefits of co-design techniques, namely rapid prototype solutions, automatic code generation, simulation, and system verification properties. In this manner, the authors propose an MDE method that employs two well-known frameworks, synchronous dataflow and place/transition Petri nets.

In [2], the authors proposed a methodology aiming at studying and analyzing the storage resources (in general) of synchronous dataflow and Petri nets using Petri net verification techniques. In this paper, the authors predicted the necessary amount of storage resource for each arc, thereby revealing the potential and effective maximum number of tokens for each synchronous dataflow without any energy consumption or optimization awareness.

To the best of the authors' knowledge, the issue of start-up conditions with SDF is addressed in [3] with integer linear programming aiming at minimizing the amount of data in the channels.

Furthermore, many researchers have devoted their efforts throughout the years to Petri net in the manufacturing and scheduling of discrete event systems. In [4] James et al. provided a colored timed Petri net hierarchical scheme for the scheduling and control level of flexible manufacturing cells. MengChu Zhou et al. in [5] investigated Petri nets in semiconductor manufacturing systems. A case study of an integrated circuit wafer fabrication system was addressed to the merging of submodules, ending with performance evaluation using a simulation tool. Gonca et al. [6] reviewed the applications of the Petri net in production scheduling in conjunction with other methods, namely with heuristic dispatching rules, search algorithms, mathematical programming approaches, and meta-heuristics approaches. Shuang Cang et al. [7] explored several contributions engaging Petri nets and artificial intelligence search in Flexible Manufacturing Systems (FMS). Flexible manufacturing scheduling problems with blocking constraints were studied in [8], supported by a Petri net approach. Herein, the authors proposes an algorithm that joins a modified A* search algorithm (beam A* search) and a node pruning strategy, to improve performance and guarantee deadlock-free schedules in FMS with blocking. Hyun-Jung Kim et al. [9] focused their attention on non-cyclic scheduling on a timed Petri net supported by a branch and bound heuristic rule. Non-cyclic scheduling problems can be found in resource-constrained project scheduling problems or robotized manufacturing systems. The branch and bound approach proved its effectiveness in a single-armed cluster tool when compared with a general mixed integer programming heuristic. In [10], Drakaki et al. used Timed Colored Petri Nets (TCPN) and reinforcement learning to model a manufacturing system and to implement the scheduling, in a way to respond to and accommodate environmental changes efficiently. Also under the framework of TCPN, [11] implemented a simulation experiment of a self-adaptive collaboration of production-logistics systems, aiming at validating the performance and the applicability of the proposed method. Davidrajuh in [12] proposed a methodology referred to as activity-oriented Petri net to favor and ameliorate the modeling of resource scheduling in discrete event dynamic systems even if the analyzed system is large and complex.

More recently, some authors devoted their attention to other Petri net applications, such as fault diagnosis and restoration for power transmission systems [13], modeling and race detection of ladder diagrams [14], and the one-wafer cyclic scheduling of treelike hybrid multi-cluster tools [15]. Likewise, some later work concerning dataflows and state-space reduction were published recently. Liu et al. [16] devoted their attention to the soundness of a workflow system. Wang et al. [17] studied the behavior consistency computation for workflow nets with unknown correspondence and to alleviate the state explosion. Xiang et al. [18] proposed the unfolding technique of Petri nets.

The authors are aware of the relevance of using Petri nets to provide both formal qualitative and quantitative analyses (performance properties), but this work aims at a qualitative analysis focused on choosing the correct start-up conditions for streaming data applications modeled by the synchronous

dataflow model. Thus, the paper does not present a formal approach for performing quantitative analysis, which would require timed or time Petri nets for computing metrics. Concomitantly, the employed mapping rules are currently restricted to place/transition Petri nets. However, with the methodology proposed in this paper, one can study different firing strategies that may also embrace multi-processor SDF architectures. A comparative study with another method has already been performed by the authors; more specifically, it was shown that the other method was not the best one to investigate the buffer memory requirements, which can be seen in [19]; thus, this paper starts from other published work.

The main contributions of this paper are the following: (i) estimating initial markings for streaming applications supported by SDF graphs that allow one to know at design time the minimum total amount of data buffer memory required in the start-up phase; (ii) proposing the use of the stubborn set method as a reduced state-space to ease the exploitation of the equivalent state-space in the Petri net framework; (iii) enumerating and discovering the possible static schedule in an SDF graph supported by an algorithm for finding all the elementary circuits adapted from [20]; and (iv) discussing the impact of distinct firing semantics in the Petri net framework on the behaviors of signal processing applications.

This document is structured into eight sections as follows. At the beginning, a small summary of the two main formalisms employed in this paper, place/transition Petri nets and dataflow, is presented. Place/transition Petri nets and the invariant method are then discussed in Section 3. In Section 4, the outlined framework of the proposed mapping rules is briefly summarized, followed by the proposed mapping rules between dataflow and Petri nets to establish connection links. The experimental results are presented in Section 5, with particular focus on the determination of the model initial conditions, which is supported by Petri net mathematical formalisms and the chosen scheme used to search for static schedules in SDFs. Two SDF models in the signal processing field are discussed. Section 6 describes the tools and computational environments used to perform the experiments. At the end of the paper, in Section 7, some conclusions are drawn, and future and on-going work are mentioned.

## 2. Petri Nets and Dataflow Models

This section presents a brief overview of the two main modeling formalisms used in this paper, Petri nets and dataflow, including the basic notation. In addition, the generic concepts, properties, and analysis techniques addressed in both fields are discussed.

Using high-level models, such as dataflow or Petri nets, enables the designer to express a design without any compromise in terms of either hardware or software. Moreover, this representation allows the designer to gain further insight about the specification and decide what is the optimal implementation choice.

### 2.1. Petri Nets

Petri nets are one of the most well-established modeling techniques. They are a graphical and mathematical modeling tool. In the past few decades, several special classes have been developed to support specific application fields (such as workflows [21] or for developing a decision process [22]), which culminated in some cases in the development of several software tools, such as TINA (TIme petri Net Analyzer) [23], SESA [24], GreatSPN (Generalized Stochastic Petri Nets) [25], CPN (Colored Petri Net) Tools [26], and IOPT (Input Output Place Transitions Nets)-Tools [27,28], just to name a few very successful examples from the academic context. To obtain a picture of the multitude of oriented-based Petri net software tools developed so far, please refer to Petri Nets Tools Database Quick Overview (https://www.informatik.uni-hamburg.de/TGI/PetriNets/tools/quick.html).

A Petri Net (PN) is a special bipartite directed graph, whose vertices (nodes) can be divided into two disjoint and independent sets, the places and the transitions [29,30], with an initial marking, which is usually known as the initial state of the model.

A marking, represented by black dots or numbers in places in a PN, is always a non-negative quantity (integer) and may represent the value of a condition or an object. The former marking represents a special arrangement of the tokens defining the initial state of the modeled system. By definition, one can find two types of nodes, places and transitions. Places, which are represented by circles or ellipses, model conditions or objects, whereas transitions are depicted by two forms, bars or rectangular boxes. Transitions are used to describe system activities, such as events or actions, that may change the system state. Places and transitions are linked by directed arcs. A place in a Petri net is commonly associated with buffers and registers for keeping information. The knowledge that a net is bounded or safe (irrespective of the assumed firing sequence) assures that no overflows occur in those buffers or registers. Moreover, there can be multiple arcs between places and transitions, which are labeled with numbers (near the arcs) to specify their multiplicity.

An example of a Petri net is shown in Figure 1a. This model has six places ($p_0...p_5$) and four transitions ($t_0...t_3$), which is a behaviorally equivalent Petri net model of the MP3 (Motion Picture Experts Group Layer-3) playback application dataflow model (Figure 1b), which is used in Section 5 as the second case study.

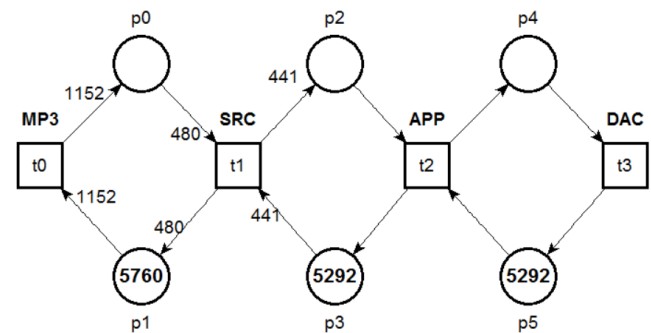

(**a**) A place/transition Petri net model

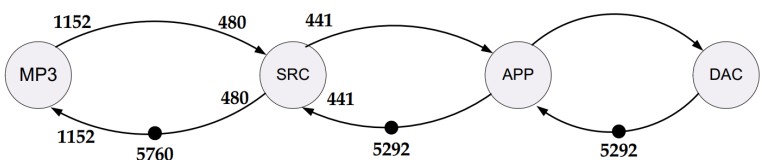

(**b**) A synchronous dataflow model

**Figure 1.** Models of computation. SRC, Sample Rate Converter; APP, Audio Post-Processing.

A transition is enabled and may fire in a marking sequence if the input place(s) ensures at least as many tokens as the multiplicity of the input arcs. Taking for example Figure 1a, only transition $t_0$ is enabled, such that the fulfillment of this previous constraint is ensured. Enabled transitions can fire anytime, discarding tokens from their input places according to the input arcs' multiplicity and adding tokens at their output places under the weight of the arcs linking the transition and their output places. Departing from an initial marking, the firing of a transition might result in new markings, which in turn may result in new transition firings, which can lead to an endless process. Moreover, one can adopt one of two firing semantics [23]: (1) interleaved semantics, in which in every marking, all possible sets or subsets of enabled transitions are fired as long as they are not in conflict; and (2) maximal step semantics, in which in each firing step, a maximal set of concurrent fireable transitions is allowed to fire. IOPT (Input Output Place Transitions Nets)-Tools adopted this type of firing semantics, thereby allowing analysis and study based on more encompassing state-spaces.

As stated above, given an initial marking $M_0$, new markings are achieved by firing enabling transitions, and if $M$ depicts a new marking attained by a firing sequence, the PN driving activity

mechanism can be described in the form of an algebraic Equation (1), which is known as a PN state equation:

$$M = M_0 + W \cdot f \tag{1}$$

where $W$ is the incidence matrix with dimension $m \times n$, in which $m$ stands for the number of places and $n$ the number of transitions, and $f$ expresses the firing vector. Each entry in the incidence matrix reflects a balance between tokens, which may be positive, negative, or zero if there is no token change involving some transition and place in the regarded Petri net. A positive balance indicates a token gain, whereas a negative balance corresponds to a token loss.

## 2.2. Dataflows

In the 1970s, Gilles Kahn developed the principles of a language for parallel programming. The seminal paper [31] explored the syntax, semantics, and the graphical representation of simple processes. The central key point lies in the translation of programs using two elements, nodes and arcs. Nodes or actors account for calculus operations, whereas edges represent the stream of data acting like First In, First Out (FIFO) lines to hold back data amounts embodied in items usually referred to as tokens. In this context, nodes play the role of processes that are interconnected by arcs representing data streams, i.e., one-way communication flow channels in which write/read operations are non-locking/locking operations. Figure 1b illustrates a dataflow model.

Kahn process networks are characterized by their monotonic and continuous properties. Monotonicity implies that higher-volume input streams of data at the nodes will generate more output information flow. This is a key property once it enables parallel computation, whereas the continuity property states that a continuous increase in a dataflow chain is translated into another increasing dataflow chain.

Synchronous Dataflow (SDF) is an extension of dataflow in which the number of data samples is known before runtime, i.e., at compile time. In [32], Lee and Messerschmitt noted requirements for exactness of the SDF for uniform parallel processors sharing memory. Under this standard, calculation procedures are engendered with nodes and arcs. Nodes embody computations, and arcs are linked to information roadmaps. In turn, nodes might not have incoming arcs. Given this situation, the program execution may be triggered at any given time. Furthermore, the availability of data on its incoming or outgoing datapath arcs may trigger nodes suddenly.

SDF also uses two types of nodes, synchronous and asynchronous nodes. Asynchronous nodes are helpful for conditional execution of subgraphs resembling the programming statement if-then-else. SDFs are appropriate for dataflows in such a manner that the engaged sample rates are rational multiples of other sample rates. In synchronous dataflow, the number of tokens generated and spent is constant, which enables employing static schedules.

The choice of synchronous dataflow is manifold and above all supported by the following: (1) it is a very mature model of computation; (2) it is suitable for streaming audio/video signal-processing applications; and (3) the data items' production and consumption rates are well known at compile time, i.e., the amounts of data produced and consumed in SDFs are fixed for each channel, thus resulting in predictable and decidable optimization problems.

Figure 2 shows an example synchronous dataflow, in which a delay is symbolized by a diamond and a number (delay amount) placed inside the symbol. This SDF processes information in an endless and cyclic manner since actor M is capable of producing streams of data at any time, whereas actor P only consumes delayed data produced by actor N.

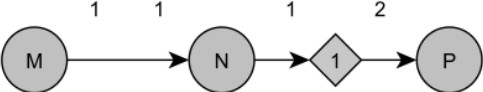

**Figure 2.** Example of a Synchronous Dataflow (SDF) with one initial delay among actors N and P (adapted from [33]).

## 3. Place/Transition Petri Nets and the Invariant Method

At the design stage phase, practitioners increasingly emphasize system analysis of qualitative and quantitative issues and currently are ruled by the pressure of the increasingly short time-to-market. Petri nets are indeed a powerful tool that can support both qualitative and quantitative analyses. The properties of a Petri net can be observed as a two-fold goal design level; a Petri net (1) allows for validating the model and (2) checking the major relevant system properties. In this manner, two types of analysis can be performed in the Petri net domain: qualitative and quantitative analyses.

To study the properties of the obtained Petri net models in a qualitative manner, the outlined approach is based on T-invariants and P-invariants. Still on the side of qualitative properties, one finds elementary properties that depend only on the arrangement of places, transitions, and arc connections; such properties are dependent on the marking featured in the Petri net structure.

The focus of this paper is on the basic place transition Petri nets, primarily owing to the strength of their analytical properties. Since the basic constructs of every Petri net are places and transitions, one can find persistency over a set or subset of places or transitions. In the first case (P-invariants), the sum of the tokens remains unchanged, whereas in the second (T-invariants), a transition firing sequence evolves the system back into the first initial marking. Mathematically, P- and T-invariants are both integer solutions of a homogeneous equation derived from the fundamental state Equation (1) for a Petri net. In both situations, the set of all nonzero entries is called the support of an invariant. Therefore, given a support, it is possible to draw a unique minimal invariant, known as the minimal support invariant ($S_I$), for each and every set (subset) of places or transitions, if it exists. With the set of all attainable minimal support invariants, one can generate a multitude of invariants, that is any invariant can be written as a linear combination of the minimal support invariants [34]. This process yields the set of all possible linear combinations of vectors, the span of $S_I$, which leads to the following: (i) any P-invariant is a linear combination of the minimal support P-invariants, $S_{PI}$; and (ii) any T-invariant is a linear combination of the minimal support T-invariants, $S_{TI}$. From this stage, one can state that (ii) leads to live Petri net models and additionally to a set (or subset) of initial markings for which the state-space is minimal, which is shown in Section 5, thus easing the state-space structure analysis. For further reading about this topic, please see [29,35–38].

Invariants in the Petri net domain mean that certain conditions/states hold even after a sequence of events or event. Petri net invariants can be decomposed into place and transition invariants. With P-invariants, one can account for places wherever the number of tokens remains constant; consequently, each place in a P-invariant is bounded. Likewise, with T-invariants, it is possible to foresee the firing sequence of transitions that returns to an initial state.

In conclusion, the key factor for the several contributions outlined in this paper in Section 1 is a Petri net invariant analysis.

## 4. Proposed Mapping Rules

It is well known that in the signal-processing field, one can establish a mapping between the time and frequency domains since signals and systems have time and frequency representations. Moreover, analyzing signals in the time domain may prevent important signal issues. It is thus required and essential to establish a "two-way bridge" between these two domains (time and frequency). Sometimes, problems are readily fixed in the time domain, whereas others are better addressed in the frequency domain; analysis of the system's behavior is generally more convenient and more intuitive in the

former domain. This "two-way bridge" mathematical relationship can be well illustrated by the well-known Fourier transform.

Some translation techniques (wavelet-based techniques) are used to ameliorate the system's inspection processes in identifying and evaluating internal defects in castings [39] and to improve machine fault diagnostic analysis [40]; extensions (e.g., Hough transform-based methods) of existing transforms that proved inadequate due to time-varying statistical properties in the signal/system, as in [41]. The above are only a few examples from the signal-processing field.

The connection between the two main domain frameworks (Petri nets and dataflow models), which was discussed previously in Section 2, is addressed using a similar underlying idea, to strengthen, highlight, and ultimately improve the dataflow framework model analysis.

The proposed mapping, shown in Figure 3, is supported by a proven natural correlation founded in both structural element domains. In each domain, one can distinguish elements that are inherently very active due to their nature and others with no embedded activity at all. In this manner, a comparison and a mapping can be achieved. A dataflow comprises processes or functions (active embedded elements) connected by arcs (non-active elements), whereas a place-transition Petri net includes transitions (active embedded elements) and places (non-active elements). This mapping was somewhat discussed in a preliminary and not in-depth manner in [42] and is devoted to timed Petri net models, to study performance issues, namely the time-optimal loop schedule in polynomial time.

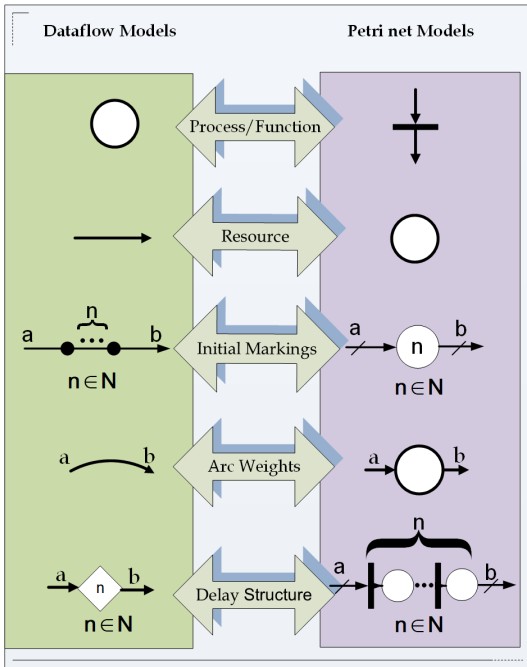

**Figure 3.** Proposed mapping rules between the engaged model domains (adapted from [43]).

On the left side of Figure 3, one finds the dataflow components, whereas on the right side, the matching Petri net components are presented. Figure 3 presents the possible liaisons established between the two engaged domains. Herein, at the top, each transition in the Petri net domain is modeled by a function in the dataflow domain. A place (resource) has as an opposing element on an arc on the dataflow side. Initial markings can also be assigned to resources by adding inscriptions or black dots to them (tokens), whereupon in Figure 3, *n* belongs to the set of natural numbers ($n \in N$). Additionally, arc weights symbolizing consumption/production token rates can be attached in both domains. If an SDF comprises arcs with initial delays, one must also to model this type of structure in the Petri net model; therefore, each and every SDF delay structure has a corresponding Petri net structure, as presented at the bottom of Figure 3. However, it must be stressed that arcs with delays are

not considered in this study, since the paper focus is on a qualitative analysis, which does not require timed or time Petri nets for computing metrics.

In this manner, it is possible to establish interoperability between these two engaged domains. Moreover, this interrelated concern needs to be biunivocal and bidirectional, thus allowing one to also make conclusions about dataflow models. This translation allows one to make conclusions about the following: (1) the necessary amount of storage resources for each arc (i.e., each resource); (2) the overall memory requirements for any signal-processing application (such as a spectrum analyzer or an MP3 playback system); (3) the minimum amount of memory requested for the start-up phase, in addition to the exact amount of required memory to perform static scheduling strategies; (4) the relevance of choosing the correct initial markings (initial resources); and (5) which T-invariants map to the static scheduling (fixed execution cycle) in an SDF framework.

**Definition 1.** *Reversibility rule. Each PN model (given the proposed mapping rules of Figure 3) of a chain order can be made reversible as long as it is possible to augment the former equivalent net by adding extra Petri net elements, taking into account the starting and ending new transitions and keeping initial markings $M_0$.*

Furthermore, based on the translation mechanisms, the following proposition can be addressed:

**Proposition 1.** *Every dataflow composed by a chain/order of actors is mapped into a PN based on a set of mapping rules outlined in Figure 3.*

Definition 1 is a key requirement to guarantee the existence of P- and T-invariants during invariant analysis (structural analysis) and a sufficient condition to ensure liveness. Moreover, in practice, it is required that every signal processing system exhibits cyclic behavior. For example, in [44], by adding a monitor place in flexible manufacturing systems modeled by a Petri net, a non-max marked siphon (A siphon is a subset of places that any input transition of a place is an output transition of some other place. It means that if a siphon has no token at some marking due to system activity, it will remain without tokens forever; consequently, that part of the system will be non-deadlock-free.) is iteratively converted into a max-marked siphon. Furthermore, regarding initial markings, in [45], Ding Liu et al. proposed a method to set the proper configuration for a Petri net system supported by the framework of [44].

## 5. Experimental Results

A basic SDF is first analyzed and discussed in Section 5.1 to introduce the proposed mapping between the two engaged frameworks, thus unveiling the behavior of the system under the possible transition firing semantics, in addition to the foreseen static schedules, and to present the impact of choosing the correct initial markings in an SDF.

To address a real case, Section 5.2 presents and discusses a signal-processing application to determine the correct initial markings and the application static schedule by inspecting the reduced state-space obtained via the stubborn set method [46].

### 5.1. Case Study I: A Basic Synchronous Dataflow with a Delay Unit

As a first example, consider the SDF model from Figure 2. This dataflow has three actors (M, N, and P) and two arcs (buffers), in which an arc has one delay unit attached to it. If a dataflow graph includes delays, an equivalent Petri net delay structure must be inserted in the final Petri net model. In this context, considering the dataflow in Figure 2 and using the delay structure illustrated in Figure 3 jointly with the proposed mapping mentioned in Section 4, an equivalent behaviorally Petri net model was obtained, as stated by Proposition 1; this model is depicted in Figure 4. However, the model shown in Figure 4 entails more than following the mapping rules between the two frameworks to achieve an equivalent Petri net model; rather, it is also necessary to exhibit a cyclical behavior, as in the

departure dataflow model. In this manner, in each PN model, it is mandatory to include place $p_2$ for the reversibility rule established by Definition 1 to hold. It should be clarified that Petri nets do not need to be cyclic, but in this case, what is required is to find static schedules; therefore, they should be cyclic.

The Petri net model shown in Figure 2 will be used as a first and simple example to illustrate the key points about the system's start-up conditions and their relevance for the system's evolution.

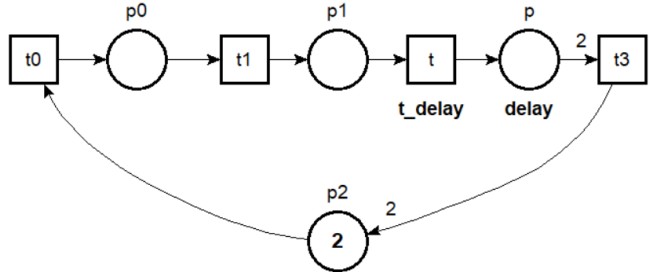

**Figure 4.** Augmented Petri net model associated with the SDF of Figure 2, using the reversibility rule established by Definition 1.

By finding each and every possible elementary cycle in the state-space, one has the possibility of determining whether the system under analysis can reach a state/event as a result of a prerequisite functional behavior or even know in advance if a firing sequence of a required functional behavior was achieved.

When a transition fires, two approaches or interpretations may be considered: (1) single-server semantics, in which it is always assumed that a transition cannot be fired more than once at a time, even if enabled to fire multiple times; and (2) infinite server semantics, in which enabled transition can be fired multiple times at once [47]. The firing semantics used in this case (Figure 4) is the single-server semantics.

In the Petri net shown in Figure 4, at the start, only state $p_2$ is marked with two tokens ($M_0 = [p_0\ p_1\ p_2\ p] = [0\ 0\ 2\ 0]$). The generated state-space is illustrated in Figure 5, in which one can see that the Petri net model has ten states. On each state-space, a depth-first search algorithm is performed to identify all elementary circuits [48] present in the Petri net model. Moreover, from P-invariants of a Petri net, one can know if the net is bounded, simply by verifying that all the elements of the P-invariants are strictly positive.

Table 1 illustrates that choosing the correct initial conditions has a major impact on the state-space dimension. For instance, if place $p_2$ has at the start four tokens, a state-space with thirty-five states and seventy transitions is engendered. The other rows of Table 1 depict several possible initial tokens in place $p_2$, which ends up with larger state-spaces. Therefore, one can conclude that even for a small dataflow, the corresponding Petri net state-space can be very large if the right initial conditions are missing or not known in advance during the design stage.

**Table 1.** Number of fired transitions and global states in the state-space as a function of the initial marking at $p_2$.

| Place $p_2$ Initial Tokens | # of Transitions | # of States |
|:---:|:---:|:---:|
| 2 | 13 | 10 |
| 4 | 70 | 35 |
| 6 | 203 | 84 |
| 8 | 444 | 165 |
| 10 | 825 | 286 |

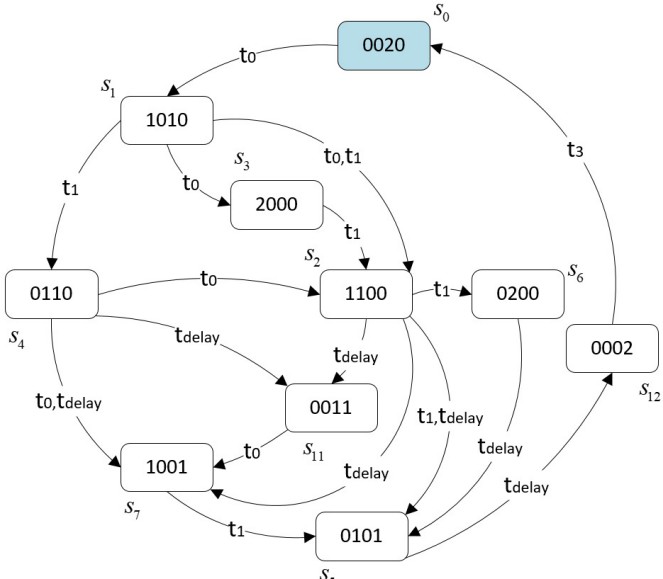

**Figure 5.** State-space (coverability tree) of the equivalent Petri net model illustrated in Figure 4, which shows all the markings that can be reached starting from an initial marking ($M_0 = [p_0 \; p_1 \; p_2 \; p] = [0\,0\,2\,0]$) and subsequently firing a sequence of enabled transitions, using single-server semantics.

To describe a static schedule, T-invariants are used, which correspond to the set of transitions that reproduce the initial marking following a suitable order. The equivalent Petri net of Figure 4 has just one T-invariant vector,

$$[2 \cdot t_0, 2 \cdot t_1, 2 \cdot t_{delay}, t_3] \qquad (2)$$

meaning that in each static schedule, transitions $t_0$, $t_1$, and $t_{delay}$ will fire twice, whereas transition $t_3$ will fire just once. This invariant yet does not specify the order of transition firings; rather, it only indicates their behavior under a periodic admissible schedule. However, to gain a better insight of how each enrolled transition is in the sequential firing and how many possible static schedules can be reached, a state-space analysis must be performed, since with T-invariants, it is only possible to know in advance how many times each transition will fire in a cycle. Therefore, if one wants to improve the knowledge of the system, it is mandatory to search the generated state-space. To account the number of static schedules in an SDF, an algorithm must be used to find all the elementary circuits in the state-space. This task allows one to find alternative schedules, which can reduce the memory storage demands. By using the Johnson algorithm [20], this task can be achieved.

Table 2 presents the eleven possible static schedules reached and found after examination of the matching generated state-space depicted in Figure 5, but only five of the eleven periodic static schedule are in agreement with T-invariant from (2), to know Cycles 1, 2, 3, 6, and 7, identified by symbol †. The representation $\lambda \cdot t_\mu$ (like for instance $2 \cdot t_{delay}$ in Table 2, Periodic Schedule (PS) Number 3) in each of the following tables (Tables 2 and 3) means that transition $t_\mu$ will fire sequentially $\lambda$ times. One can conclude that the initial markings of a Petri net will comprise the entire T-invariant space. Thus, only a few will be observable, as indicated in Table 2. Since the firing semantics used by the IOPT Tool is the maximal step, two or more concurrent fireable transitions can fire, to introduce and unveil parallel and/or distributed processing activities. Table 2 depicts manifold cases (Cycles 4, 5, 8–11), where each notation $[t_x(*), t_y(*)]$ represents a block of concurrent fired transitions $(t_x, t_y)$. Furthermore, the nodes engaged in each Periodic Admissible Static Schedule (PASS) are detected. Table 4 depicts the node sequences for each PASS, including the solutions that follow (2).

However, if the aim is to find periodic admissible sequential schedules, then only a single actor can be fired at a time, meaning that on the Petri net side, the adopted firing strategy must be interleaving semantics, instead of the maximal step. Table 3 presents five possible sequential schedules. Table 5 also specifies the node sequences enrolled in each sequential schedule. Tables 3 and 5 are as a consequence of the firing strategies subsets of Tables 2 and 4, respectively.

**Table 2.** Possible firing sequences (PS, Periodic Schedule) with an initial marking vector $[p_0 \ p_1 \ p_2 \ p] = [0 \ 0 \ 2 \ 0]$ under a maximal step firing strategy.

| # of PS | Periodic Schedule Firing Sequences |
|---|---|
| 1 [†] | $t_0, t_1, t_{delay}, t_0, t_1, t_{delay}, t_3$ |
| 2 [†] | $t_0, t_1, t_0, t_{delay}, t_1, t_{delay}, t_3$ |
| 3 [†] | $t_0, t_1, t_0, t_1, 2 \cdot t_{delay}, t_3$ |
| 4 | $t_0, t_1, t_0, [t_1(*), t_{delay}(*)], t_{delay}, t_3$ |
| 5 | $t_0, t_1, [t_0(*), t_{delay}(*)], t_1, t_{delay}, t_3$ |
| 6 [†] | $2 \cdot t_0, t_1, t_{delay}, t_1, t_{delay}, t_3$ |
| 7 [†] | $2 \cdot t_0, 2 \cdot t_1, 2 \cdot t_{delay}, t_3$ |
| 8 | $2 \cdot t_0, t_1, [t_1(*), t_{delay}(*)], t_{delay}, t_3$ |
| 9 | $t_0, [t_0(*), t_1(*)], t_{delay}, t_1, t_{delay}, t_3$ |
| 10 | $t_0, [t_0(*), t_1(*)], t_1, 2 \cdot t_{delay}, t_3$ |
| 11 | $t_0, [t_0(*), t_1(*)], [t_1(*), t_{delay}(*)], t_{delay}, t_3$ |

**Table 3.** Possible firing sequences (SS, Static Schedule) for initial marking vector $[p_0 \ p_1 \ p_2 \ p] = [0 \ 0 \ 2 \ 0]$ under single-server semantics.

| # of SS | SS Firing Sequences |
|---|---|
| 1 | $t_0, t_1, t_{delay}, t_0, t_1, t_{delay}, t_3$ |
| 2 | $t_0, t_1, t_0, t_{delay}, t_1, t_{delay}, t_3$ |
| 3 | $t_0, t_1, t_0, t_1, 2 \cdot t_{delay}, t_3$ |
| 4 | $2 \cdot t_0, t_1, t_{delay}, t_1, t_{delay}, t_3$ |
| 5 | $2 \cdot t_0, 2 \cdot t_1, 2 \cdot t_{delay}, t_3$ |

The former case study demonstrated how initial markings can be achieved and their impact when the right initial markings are not known; concomitantly, this introductory case also highlights the possibility of studying a signal-processing system in a broader sense by adopting a different firing strategy, such as the maximal step semantics instead of being tied to a more conservative approach, in which the goal is to reach periodic admissible sequential schedules (interleaving semantics).

**Table 4.** Node sequences for initial marking vector $[p_0 \ p_1 \ p_2 \ p] = [0 \ 0 \ 2 \ 0]$ under a maximal step firing strategy.

| # of PS | Node Sequences |
|---|---|
| 1 [†] | $s_0 \mapsto s_1 \mapsto s_4 \mapsto s_{11} \mapsto s_7 \mapsto s_5 \mapsto s_{12} \mapsto s_0$ |
| 2 [†] | $s_0 \mapsto s_1 \mapsto s_4 \mapsto s_2 \mapsto s_7 \mapsto s_5 \mapsto s_{12} \mapsto s_0$ |
| 3 [†] | $s_0 \mapsto s_1 \mapsto s_4 \mapsto s_2 \mapsto s_6 \mapsto s_5 \mapsto s_{12} \mapsto s_0$ |
| 4 | $s_0 \mapsto s_1 \mapsto s_4 \mapsto s_2 \mapsto s_5 \mapsto s_{12} \mapsto s_0$ |
| 5 | $s_0 \mapsto s_1 \mapsto s_4 \mapsto s_7 \mapsto s_5 \mapsto s_{12} \mapsto s_0$ |
| 6 [†] | $s_0 \mapsto s_1 \mapsto s_3 \mapsto s_2 \mapsto s_7 \mapsto s_5 \mapsto s_{12} \mapsto s_0$ |
| 7 [†] | $s_0 \mapsto s_1 \mapsto s_3 \mapsto s_2 \mapsto s_6 \mapsto s_5 \mapsto s_{12} \mapsto s_0$ |
| 8 | $s_0 \mapsto s_1 \mapsto s_3 \mapsto s_2 \mapsto s_5 \mapsto s_{12} \mapsto s_0$ |
| 9 | $s_0 \mapsto s_1 \mapsto s_2 \mapsto s_7 \mapsto s_5 \mapsto s_{12} \mapsto s_0$ |
| 10 | $s_0 \mapsto s_1 \mapsto s_2 \mapsto s_6 \mapsto s_5 \mapsto s_{12} \mapsto s_0$ |
| 11 | $s_0 \mapsto s_1 \mapsto s_2 \mapsto s_5 \mapsto s_{12} \mapsto s_0$ |

**Table 5.** Node sequences for initial marking vector $[p_0\ p_1\ p_2\ p] = [0\ 0\ 2\ 0]$ under single-server semantics.

| # of SS | Node Sequences |
|---------|----------------|
| 1 | $s_0 \mapsto s_1 \mapsto s_4 \mapsto s_{11} \mapsto s_7 \mapsto s_5 \mapsto s_{12} \mapsto s_0$ |
| 2 | $s_0 \mapsto s_1 \mapsto s_4 \mapsto s_2 \mapsto s_7 \mapsto s_5 \mapsto s_{12} \mapsto s_0$ |
| 3 | $s_0 \mapsto s_1 \mapsto s_4 \mapsto s_2 \mapsto s_6 \mapsto s_5 \mapsto s_{12} \mapsto s_0$ |
| 4 | $s_0 \mapsto s_1 \mapsto s_3 \mapsto s_2 \mapsto s_7 \mapsto s_5 \mapsto s_{12} \mapsto s_0$ |
| 5 | $s_0 \mapsto s_1 \mapsto s_3 \mapsto s_2 \mapsto s_6 \mapsto s_5 \mapsto s_{12} \mapsto s_0$ |

*5.2. Case Study II: A Signal Processing Application*

The experimental results will be supported and illustrated by a well-known model in the dataflow community, MP3 playback [49]. Figure 1b depicts the corresponding dataflow model of the MP3 playback application from [49]. In this application, compressed audio is decoded by the MP3 actor into a 48-kHz sample stream. Afterwards, these samples are converted by the Sample Rate Converter (SRC) into a 44.1-kHz stream, which is followed by the Audio Post-Processing (APP) actor to enhance the stream audio quality. Finally, the stream sample is converted back to the analog domain using a Digital-to-Analogue Converter (DAC). In spite of being a dataflow with a few actors, the demanded cycles to the MP3 playback application in every static schedule are considerable, since each actor (MP3, SRC, APP, and DAC) performs [5, 12, 5292, 5292] repetitions, respectively. Figure 1a shows the counterpart of the MP3 playback application in the Petri net domain, wherein the suitable initial markings (in places $p_1$, $p_3$, and $p_5$) were established by performing the invariant analysis, in this case the T-invariants.

Signal-processing system applications, cyber-physical systems, and industrial applications usually have large state-spaces committed to several hundred million states; therefore, it is neither desirable nor feasible to inspect them visually. Moreover, during the design process, models suffer modifications, and the state-space needs to be rechecked to ensure that everything is functioning properly, if a transition or set of transitions was fired, or if a specific net marking was reached.

In this manner, it will be possible with the proposed mapping in Section 4 to query the state-space to know if those previous requirements (Definition 1 and Proposition 1) were fulfilled, which empowers and strengthens the model-checking system. A further improvement can be achieved if a method for generating a reduced state-space is used, namely using stubborn sets [46]. This technique takes advantage of the lack of interaction between transitions while preserving their liveness.

Supported by the P-invariant analysis, one can establish that the Petri net model in Figure 1a has three invariants,

$$M(p_0) + M(p_1) \quad = \quad 5760 \tag{3}$$
$$M(p_2) + M(p_3) \quad = \quad 5292 \tag{4}$$
$$M(p_4) + M(p_5) \quad = \quad 5292. \tag{5}$$

From the previous Equations (3)–(5), one can achieve the following maximum buffer requirements for each place (their capacities) in the PN model (arc in the associated dataflow model):

$$M(p_0) \quad = \quad M(p_1) = 5760 \tag{6}$$
$$M(p_2) \quad = \quad M(p_3) = 5292 \tag{7}$$
$$M(p_4) \quad = \quad M(p_5) = 5292. \tag{8}$$

The associated T-invariant for this model is:

$$[5 \cdot t_0, 12 \cdot t_1, 5292 \cdot t_2, 5292 \cdot t_3]. \tag{9}$$

From the existence of (9), one concludes that the MP3 playback application has one static schedule in which $t_0$ fires five times, $t_1$ twelve times, and $t_2$ and $t_3$ 5292 times; moreover, $\sum(\#firings) = 10{,}601$. However, the designer is still faced with the lack of information about the transition firing order in the former static schedule. This further information can be obtained if the state-space is traversed to search for all possible schedules firing sequences founded on the T-invariant expression. Furthermore, Expression (9) allows one to know in advance the initial conditions for each set of places enrolled in P-invariant analysis (3)–(5), as follows in the set of places $(p_0, p_1)$, $(p_2, p_3)$, and $(p_4, p_5)$.

The simulation results were obtained choosing as initial marking conditions for vector marking places $[p_0 \ p_1 \ p_2 \ p_3 \ p_4 \ p_5] = [0 \ 5760 \ 0 \ 5292 \ 0 \ 5292]$, as shown in Figure 1a, with single-server firing semantics.

To minimize the state-space search effort cost, owing to the large size of the associated state-space of this data stream application, a state-space reduction technique was applied. The stubborn reduction method is a state-space reduction method that takes advantage of concurrency, or of the lack of interaction between transitions preserving the liveness of transitions and all terminal states, as well as the existence of non-termination. Therefore, applying the stubborn reduction over the state-space of the equivalent place/transition Petri net, one can achieve a very large improvement in their dimensionality. A state-space of just 15,929 states was reached, whereas with no reduction, this state-space had several hundreds of millions of states. Supported by the algorithm developed for finding cycles in the generated state-space, a cycle with 10,602 states and 10,601 transitions was found. The cycle began at Node No. 5359 (as observed below in Table 6) and evolved till State No. 15,929, after which, it returned to State No. 5329.

**Table 6.** Static Schedule (SS) for initial marking vector $[p_0 \ p_1 \ p_2 \ p_3 \ p_4 \ p_5] = [0 \ 5760 \ 0 \ 5292 \ 0 \ 5292]$ under a single-server semantics for MP3 playback application.

| # of SS | Node Sequences (Node No.) |
|---|---|
| 1 | $5329 \mapsto 5330 \mapsto \ldots \mapsto 15{,}928 \mapsto 15{,}929 \mapsto 5329$ |

Table 7 depicts in the first line the maximum capacities (A) foreseen with the support of P-invariant analysis and the following lines show the maximum capacities reached after analyzing the dynamic behavior of the system in a reduced state-space under a PASS (B) and at the start-up phase (C). Examining the state-space, the system's behavior can be split into two separate phases: (1) the start-up and (2) cyclic phase. The second line of Table 7 presents the capacity of each node for the cyclic phase, whereas the last line displays the individual node capacities for the start-up phase. This phase is indeed greedier in terms of allocated memory resources. Therefore, the system evolves through a start-up phase to traverse 5328 states. This phase can last longer if the right initial conditions are not met at the design stage. Thus, to overcome this issue, one must foresee the right initial markings in the system, precluding in this way a waste of memory resources. This task is accomplished by finding the interconnection between T-invariant and P-invariant concerning the sets of places engaged with P-invariants and applying the knowledge gained from T-invariant analysis about transition firings. The MP3 playback application took into account this former knowledge. Under a PASS, the sum of maximum required capacity for this case was $\sum_{j=0}^{5} p_j = 18{,}322$ units (last column in Table 7), whilst in the start-up phase, the sum of maximum capacities was equal to $\sum_{j=0}^{5} p_j = 32{,}688$. The start-up phase is a more demanding phase, requiring almost twice the data memory as PASS. In the presence of an overhead prologue, if one adopts a shared buffer strategy, then the maximum data memory required is 32,688 units, which is more demanding and time consuming, requiring label information, which is a penalty due to the context switching overhead.

**Table 7.** Capacities of MP3 playback application under a single-server semantics.

| Places | $p_0$ | $p_1$ | $p_2$ | $p_3$ | $p_4$ | $p_5$ | $\sum_{j=0}^{5} p_j$ |
|--------|-------|-------|-------|-------|-------|-------|----------------------|
| (A) | 5760 | 5760 | 5292 | 5292 | 5292 | 5292 | 32,688 |
| (B) | 5760 | 1536 | 5292 | 441 | 5292 | 1 | 18,322 |
| (C) | 5760 | 5760 | 5292 | 5292 | 5292 | 5292 | 32,688 |

The graph in Figure 6 depicts the profiles of storage resources allocated to each place (buffer), namely $\{p_0...p_5\}$, into two separate steps: (1) the start-up step; and (2) the static schedule step. Herein, one can follow the memory resources progress at each buffer. On the top of the graph, places $p_0$ and $p_2$ dominate and strive for higher resources in both execution steps, reaching a maximum of (5760, 5292) memory units at both phases linked to the generated state-space. Within the static schedule cycle, which started at State Number 5392, two distinct sets of places could be identified, $(p_0, p_2, p_4)$ and $(p_1, p_3, p_5)$, concerning the allocation of storage resources. The first set of places demanded more memory resources than the former set.

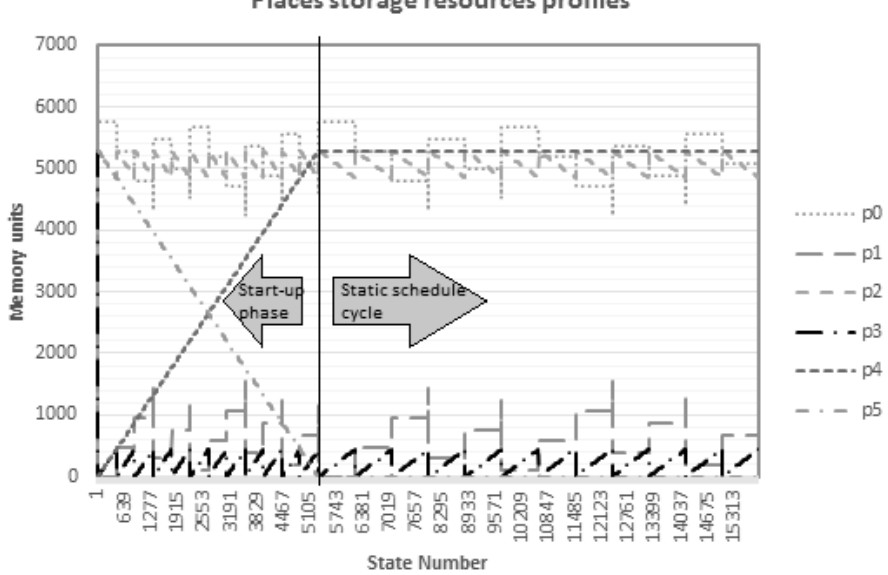

**Figure 6.** Places'/buffers' storage resource profiles.

The analysis of the state-space in addition to allowing one to know the states enrolled in the static schedule, also enables the designer to foresee the firing profile within an ongoing cycle. As mentioned previously, T-invariant analysis lays the knowledge base for the firing strategy. Thus, it is known that $t_0$ fires five times, $t_1$ twelve times, and $t_2$ and $t_3$ 5292 times, respectively. This knowledge lets one plan a methodology to reduce memory requirements along the execution cycle just as in the start-up phase.

The buffer requirement for each arc of a dataflow does not require any operational model [50], and this paper proposes a new strategy as concerns the prologue in each dataflow model. By knowing in advance the optimal initial number of tokens on arcs at the prologue, which may be shared with the buffer requirements of the scheduling list, since the prologue is executed only once, there is no need to demand for a separate data buffer as proposed in [50]. Our approach allows one to reason about the timing behavior of the system to support scheduling strategies [51] (fully-static and static order scheduling), [52].

## 6. Tools and Computational Environments

The development and analysis of the resulting Petri nets was performed on three computer-aided tools that are available on the web: (I) SESA-Signal/Net System Analyzer [53]; (II) Still TINA (TIme Petri Net) Analyzer [23]; (III) and IOPT (Input Output Place Transitions Nets) Tools [27]. The invariant analysis (used in Section 5) was likewise performed in those environments. All other experimental results' analysis based on the state-space produced by the aforementioned tools was fulfilled by software developed under GCC (GNU Compiler Collection), using C language.

## 7. Conclusions

This paper presented a semi-formal description translation mapping between SDFs and PNs that preserves behavioral semantics. This technique allows one to accommodate new trends about SDF, namely: (1) perform simulations and verify the properties of equivalent Petri net computation models; (2) make conclusions about the correct initial conditions associated with synchronous dataflow to ensure the system's liveness; and (3) reduce the runtime overhead, thus improving the system's energy consumption in the start-up phase, which can be critical in cyber-physical or real-time systems. Moreover, based on an analysis of the achieved state-space of the equivalent Petri net model, one can search every possible cycle, leading to the several potential static schedules underlying an SDF, as long as the dimensionality of the state-space is amenable. However, if instead, the state-space is not treatable, owing to its dimensionality, a reduction (using the stubborn reduction method) can be performed to yield an equivalent state-space in which the liveness and live lock of transitions still hold. This ability to analyze the behavior of models at a high level of abstraction is a major advantage for designers, whereas at the software level, for example, one must run the code to verify its correctness instead of analyzing it.

The dataflow modeling technique enables a designer to express parallel activities at the highest level of abstraction, as in Petri net modeling techniques. Such closeness as a modeling technique, among the other features described in Section 4, reinforces and strengthens the proposed methodology presented in the paper. In conclusion, this paper addressed the start-up relevance conditions of every SDF to enhance the runtime overhead and proposed the use of a reduction technique to address the state-space explosion issue, thus making the state-space more amenable in terms of its size. The paper proposed an approach to enumerate and identify the engaged static schedules in every SDF graph and discussed the impact of firing semantics.

Other enhancements may be achieved if optimization techniques such as evolutionary computation are used to minimize the data requirements among each arc in a static schedule by specifying and yielding additional constraints to buffers as much as their execution lifetime.

**Author Contributions:** Methodology, J.-I.R.; software, J.-I.R.; validation, J.-I.R.; formal analysis, J.-I.R.; investigation, J.-I.R.; resources, J.-I.R.; writing—original draft preparation, J.-I.R.; writing—review and editing, O.P.D. and L.G.; visualization, J.-I.R.; supervision, O.P.D. and L.G.; funding acquisition, L.G.

**Funding:** The APC was funded by UNINOVA, Campus de Caparica, Caparica, 2829-516 Almada, Portugal.

**Acknowledgments:** This work was partially financed by Portuguese Agency FCT (Fundação para a Ciência e Tecnologia), in the framework of Project UID/EEA/00066/2013.

**Conflicts of Interest:** The authors declare no conflict of interest.

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
