# Peer review of "Improving Synchronous Dataflow Analysis Supported by Petri Net Mappings"

_electronics, doi:10.3390/electronics7120448_

Round 1
Reviewer 1 Report
This paper proposes a model-driven engineering (MDE) method to translate a Synchronous Dataflow (SDF) into a Petri net, and utilizes P/T invariants and stubborn-set-based method to analyze its state space. The signal processing application is interesting. Some experiments are done to illustrate the effectiveness of their methods. The reviewer thinks it acceptable, but it must be improved first.
(1) The organization of this paper needs some adjustment, e.g., Part 1.1 in Section 1 can be deleted.
(2) Authors had better do some experiments to compare their method with the existing ones in order to show their method’s advantages.
(3) Authors should clarify their innovations.
(4) Some latest related work of data flows and state space reduction should be cited and reviewed, e.g.,
[A] GJ Liu, W Reisig, CJ Jiang, and MC Zhou, A Branching-process-based method to check soundness of workflow systems, IEEE Access, vol. 4, pp. 4104-4118, 2016
[B] DM Xiang, GJ Liu, CG Yan, CJ Jiang, Detecting Data Inconsistency Based on the Unfolding Technique of Petri Nets, IEEE Transactions on Industrial Informatics, vol. 13, no. 6, pp. 2995--3005, 2017
[C] M. M. Wang, G. J. Liu, P. H. Zhao, C. G. Yan, and C. J. Jiang, “Behavior consistency computation for workflow nets with unknown correspondence,” IEEE/CAA J. of Autom. Sinica, vol. 5, no. 1, pp. 281-291, Jan. 2018.
The following recent Petri net’s applications may be cited:
[D] F. J. Yang, N. Q. Wu, Y. Qiao, and R. Su, “Polynomial approach to optimal one-wafer cyclic scheduling of treelike hybrid multi-cluster tools via Petri nets,” IEEE/CAA J. of Autom. Sinica, vol. 5, no. 1, pp. 270-280, Jan. 2018.
[E] J. Luo, Q. Zhang, X. Chen, and M. Zhou, "Modeling and Race Detection of Ladder Diagrams via Ordinary Petri Nets," IEEE Transactions on Systems, Man, and Cybernetics: Systems, vol.48, No. 7, pp. 1166-1176, July 2018.
[F] Z. Jiang, Z. Li, N. Wu, and M. Zhou, “A Petri Net Approach to Fault Diagnosis and Restoration for Power Transmission Systems to Avoid the Output Interruption of Substations,” IEEE Systems Journal, 12(3), pp. 2566 – 2576, Sept. 2018.
(5) The last line of Page 13: lay --> lays.
Reviewer 2 Report
This paper presents a sound application of Petri nets to the evaluation of the needs, in terms of resources, of an application, by exploiting Petri nets invariants. The paper is well written and organized, related works are sufficient to give the readers a good introduction to the topic. The method is well explained and quite clear even for non specialist readers. Results are sound and the reader is guided through the steps of the method towards the analysis of a literature case study.
Please clarify the sentence at row 1066.
Please give a more extended description and justification of metrics at row 249-252, as in this point of the paper their relation with the model is not yet clear to non specialist readers.
Please clarify for non experts why the Petri net needs to be cyclic (it is evident to PN users, but it is not for practitioners).
At row 338, probably "its impact" should be "their impact".
In Subsection 4.2, please give a wider description of Figure 1 motivating the numbers with respect to the application and stress the binding with the application needs.
At row 353, probably "cyber-physics" should be "cyber-physical".
In Figure 3, range of variable "n" should in my opinion better given as a set, as values are integer, rather than as an interval, and please consider if it is the case to substitute the infinite symbol with an arbitrary integer bound, as "n" is the number of structural elements of Dataflow models and Petri net models, that are usually finite.
Reviewer 3 Report
The authors have proposed a novel approach to estimate whether an SDF has a static schedule by performing simulation and property verification of the equivalent based P/T Petri net system and illustrated the use of a reduction technique to address the state-space explosion issue.
I have the following comments for the authors:
1) In lines 45-64 a literature review related to PN in manufacturing and scheduling of DES has been provided. However, there are more recent studies in the literature that could be included such as:
- A study on manufacturing scheduling using Colored Petri Nets and Artificial Intelligence for FMS:
"Manufacturing Scheduling Using Colored Petri Netsand Reinforcement Learning", by Drakaki and Tzionas, Applied Sciences, 2017
- A study on production-logistics systems collaboration using Colored Petri Nets:
"A Timed Colored Petri Net Simulation-Based Self-Adaptive Collaboration Method for Production-Logistics Systems", by Guo et al., Applied Sciences, 2017.
2) Line 114: The phrase "Places and transitions are linked by one-way directed arcs" should be replaced by: "Places and transitions are linked by directed arcs".
3) Lines 191-193: The authors should clarify what they mean by: first case and second case.
4) Line 211: The phrase: "for the several contributions" should be rephrased (is not specific).
5) Line 227: The phrase: "main domain frameworks" should be rephrased, to explain which are the two frameworks (PNs and SDF?)
6) Lines 244-245: Since authors use arcs with delays they should comment in Section 3 again why they do not use timed PNs.
7) Line 359: The phrase "previous requirements" is not clear (which are the previous requirements?) because it refers to the previous paragraph.
8) Lines 377-384: The presented results should be explained more. E.g. by explaining the stubborn reduction method in general.
9) Line 431: The phrase "This paper reviewed" should be rephrased "This paper presented".
10) Lines 432-437: It is a long sentence.
11) Line 440: The reduction method should be mentioned explicily, i.e. as stubborn reduction method.
